

# Fuel trait effects on flammability of native and invasive alien shrubs in coastal fynbos and thicket (Cape Floristic Region)

Tineke Kraaij[1,2,*], Samukelisiwe T. Msweli[1,2,*] and Alastair J. Potts[2,3]

[1] Natural Resource Science and Management Cluster, Nelson Mandela University, George, Western Cape, South Africa
[2] African Centre for Coastal Palaeoscience, Nelson Mandela University, Port Elizabeth, Eastern Cape, South Africa
[3] Botany Department, Nelson Mandela University, Port Elizabeth, Eastern Cape, South Africa
* These authors contributed equally to this work.

Corresponding author
Tineke Kraaij,
tineke.kraaij@mandela.ac.za

## ABSTRACT

In June 2017, extreme fires along the southern Cape coast of South Africa burnt native fynbos and thicket vegetation and caused extensive damage to plantations and residential properties. Invasive alien plants (IAPs) occur commonly in the area and were thought to have changed the behaviour of these fires through their modification of fuel properties relative to that of native vegetation. This study experimentally compared various measures of flammability across groups of native and alien invasive shrub species in relation to their fuel traits. Live plant shoots of 30 species (10 species each of native fynbos, native thicket, and IAPs) were sampled to measure live fuel moisture, dry biomass, fuel bed porosity and the proportions of fine-, coarse- and dead fuels. These shoots were burnt experimentally, and flammability measured in terms of maximum temperature (combustibility), completeness of burn (consumability), and time-to-ignition (ignitability). Multiple regression models were used to assess the relationships between flammability responses and fuel traits, while the Kruskal-Wallis $H$ test was used to establish if differences existed in flammability measures and fuel traits among the vegetation groups. Dry biomass significantly enhanced, while live fuel moisture significantly reduced, maximum temperature, whereas the proportion of fine fuels significantly increased completeness of burn. Unlike other similar studies, the proportion of dead fuels and fuel bed porosity were not retained by any of the models to account for variation in flammability. Species of fynbos and IAPs generally exhibited greater flammability in the form of higher completeness of burn and more rapid ignition than species of thicket. Little distinction in flammability and fuel traits could be made between species of fynbos and IAPs, except that fynbos species had a greater proportion of fine fuels. Thicket species had higher proportions of coarse fuels and greater dry biomass (~fuel loading) than species of fynbos and IAPs. Live fuel moisture did not differ among the vegetation groups, contrary to the literature often ascribing variation in flammability to fuel moisture differences. The fuel traits investigated only explained 21–53% of the variation in flammability and large variation was evident among species within vegetation groups suggesting that species-specific and *in situ* community-level investigations are warranted, particularly in regard fuel moisture and chemical contents.

## INTRODUCTION

Flammability is the ability of vegetation (fuel) to burn (*Fernandes & Cruz, 2012*; *Gill & Zylstra, 2005*) and is a measure of fire behaviour relevant to studies of fire risk and fire ecology (*Keeley, 2009*). Flammability can be assessed at the scale of the vegetation community, the individual plant, or plant components in terms of ignitability (time-to-ignition, or ignition frequency), combustibility (maximum temperature, or heat released, and rate of burn), sustainability (burn duration), and consumability (completeness of burn, or biomass consumed) (*Anderson, 1970*; *Santacruz-García et al., 2019*). These measures of flammability are affected by fuel traits, weather conditions and their interactions (*Bianchi & Defossé, 2015*; *Santana, Baeza & Vallejo, 2011*; *Saura-Mas et al., 2010*). Fuel traits, through their effects on flammability and fire behaviour, ultimately have implications for fire risk management (*Bond & Midgley, 1995*; *Cowan & Ackerly, 2010*; *Fernandes & Cruz, 2012*).

Fuel traits that have relevance for flammability include fuel moisture content (or the inverse, fuel dry matter content), carbon compounds (cellulose, hemicellulose, and lignin), and volatile organic compounds (terpenes) (*Behm et al., 2004*; *Midgley, 2013*; *White & Zipperer, 2010*), and traits that relate to the structural form of fuels such as fuel size, fuel loading, bulk density, and fuel bed porosity (fuel sparseness; or the inverse, packing ratio) (*Burger & Bond, 2015*; *Saura-Mas et al., 2010*; *Viegas, 2006*). Flammability has been experimentally assessed in relation to fuel traits in several vegetation types of the world (*Cowan & Ackerly, 2010*; *Cui et al., 2020*; *Saura-Mas et al., 2010*; *Scarff & Westoby, 2006*; *Schwilk, 2003*). Such experiments have commonly considered the flammability of leaf litter (*e.g.*, *Burton et al., 2021*) or plant shoots (summarised in Supplemental S1), the latter being the focus of our study concerning evergreen shrublands that sustain canopy fires. Although burning of small plant components does not adequately represent whole plant- or community-level flammability (*Fernandes & Cruz, 2012*; *Schwilk, 2015*), experimental approaches are useful first steps to understand concepts of flammability (*Pausas & Moreira, 2012*). *Alam et al. (2020)* demonstrated that measurements of leaf flammability are decoupled from shoot flammability, and that the latter is better correlated with in-field expert observations of whole plant flammability. Generalisation of results from flammability studies is furthermore complicated by their use of different measurement methods and diverse expressions of flammability (*Alam et al., 2020*; *Burton et al., 2021*; *Schwilk, 2015*; *Wyse et al., 2016*). Many studies have expressed flammability as a composite measure (*i.e.*, a flammability index) (*Alessio et al., 2008*; *Burger & Bond, 2015*; *Calitz, Potts & Cowling, 2015*; *Santana & Marrs, 2014*) or have concentrated on singular measures of flammability or a select few fuel traits (Supplemental S1). These discrepancies along with experimentation at different scales, or only indirect investigations of the relationships

between fuel traits and flammability (Supplemental S1), render interpretation and extrapolation of these relationships challenging.

Nonetheless, trends that commonly emerged from the literature include that fuel loading increases flammability through increasing combustibility (*Baeza et al., 2002*; *Keeley, 2009*; *Saura-Mas et al., 2010*; *Simpson et al., 2016*). Fuel moisture generally depresses ignitability and the rate of fire spread (*Alam et al., 2020*; *Alessio et al., 2008*; *Bianchi & Defossé, 2015*; *Davies & Legg, 2011*; *Dimitrakopoulos, 2001*; *Murray, Hardstaff & Phillips, 2013*; *Saura-Mas et al., 2010*; *Simpson et al., 2016*) as more energy is required to heat moister fuels to combustion (*Davies & Nafus, 2013*; *Kane & Nuria, 2019*). The proportions of fine and dead fuels (in live plant shoots) consistently enhance flammability by increasing ignitability and consumability but not sustainability (*Burger & Bond, 2015*; *Calitz, Potts & Cowling, 2015*; *Davies & Legg, 2011*; *Santana & Marrs, 2014*; *Schwilk, 2003*; Supplemental S1), while coarse fuels reduce ignitability (*Alam et al., 2020*). Fuel bed porosity shows inconsistent effects on flammability in different vegetation types (*Burger & Bond, 2015*; *Davies & Legg, 2011*; *van Wilgen, Higgins & Bellstedt, 1990*; *Ward et al., 1996*).

Fuel properties of native vegetation may be modified by the presence of invasive alien plants (IAPs) with knock-on effects on flammability and fire behaviour (*Brooks et al., 2004*; *Cubino et al., 2018*; *van Wilgen & Richardson, 1985*; *Wyse, Perry & Curran, 2018*). Fuels of some IAPs contain volatile substances supporting fires of higher intensities (*Alessio et al., 2008*; *Behm et al., 2004*; *Wyse, Perry & Curran, 2018*). Rapid plant growth associated with IAPs may also result in accumulation of excessive fuel loads (*Brooks et al., 2004*; *Davies & Nafus, 2013*; *Richardson & van Wilgen, 2004*). IAPs may furthermore modify the moisture content of fuels (*Murray, Hardstaff & Phillips, 2013*), or the arrangement of fuel particles compared to that of the native vegetation (*Brooks et al., 2004*; *Murray, Hardstaff & Phillips, 2013*), for example, the vertical or horizontal continuity of fuels or the fuel bed porosity, thereby increasing or decreasing the flammability of vegetation (*Cubino et al., 2018*; *Cui et al., 2020*; *Davies & Nafus, 2013*; *Richardson & van Wilgen, 2004*). The modification of fuel traits by IAPs may ultimately lead to additional ecological impacts if the native vegetation responds to altered fire regimes (*Brooks et al., 2004*).

In June 2017, extreme fires along the southern Cape coast of South Africa burnt native fynbos and thicket shrublands and caused extensive damage to timber plantations and residential properties. After these fires, speculation was rife regarding the influence of different vegetation types, and IAPs in particular, on the severity of these fires (*e.g.*, *De Ronde, 2017*). Woody IAPs commonly occur in the area and a study using satellite-derived measures to assess variation in the severity of these fires at a landscape scale showed that biomass consumption (and thus burn severity) was higher in vegetation in which invasive alien trees and shrubs dominated than in the native fynbos and thicket vegetation (*Kraaij et al., 2018*); also, that the burn severity in thicket, which is considered a fire-resistant vegetation (*Geldenhuys, 1994*), was higher than in fynbos, which is considered fire-prone and more flammable (*Calitz, Potts & Cowling, 2015*). Both fynbos and thicket burnt more completely than the alien plant dominated vegetation with implications for subsequent fuel availability and fire hazard (*Kraaij et al., 2018*). Although *Kraaij et al. (2018)*

demonstrated discrepancies in burn severity among the vegetation types in the region, mechanistic understanding of differences between common invasive and indigenous woody plants' fuel traits and various components of flammability is lacking.

Here we extended the investigation we conducted in *Msweli et al. (2020)* to assess flammability in relation to fuel traits across species from three vegetation groups, namely IAPs, native fynbos and thicket, that are common to the coastal parts of the Cape Floristic Region of South Africa. *Msweli et al. (2020)* analysed flammability in relation to 21 sampling days spanning a wide range of fire danger conditions, whereas here we used the average of these values to explore relationships amongst flammability measures and plant traits. Flammability measures considered were combustibility (maximum temperature), consumability (completeness of burn), and ignitability (time-to-ignition), whilst the fuel traits considered were live fuel moisture, dry biomass, fuel bed porosity and the proportions of fine-, coarse- and dead fuels. Large plant shoots were used to facilitate a more realistic assessment of canopy flammability. Ultimately, our aim in this study is to improve the mechanistic understanding of the relationship between fuel traits and flammability responses in temperate shrubland species.

## MATERIALS AND METHODS

### Study region

The study region occurs along the southern Cape coast of South Africa, close to the city of George (33.964°S, 22.534°E), within the Cape Floristic Region. The climate is moderated by the maritime influence with average minimum and maximum temperatures ranging from 7–19 °C in June and 15–26 °C in January, and annual average rainfall of approximately 800 mm distributed throughout the year (*Bond, 1981*). The area experiences weather conditions suitable for fires at any time of the year and fires are often associated with hot, dry katabatic ('berg') winds (*Kraaij, Cowling & van Wilgen, 2013*; *van Wilgen, 1984*).

The vegetation of the study region is classified as Southern Cape Dune Fynbos which consists of medium-dense sclerophyllous fynbos (~fine-leaved) shrublands up to 2 m in height, interspersed with dense clumps of subtropical mesophyllous thicket shrubs or trees up to 4 m in height (*Mucina & Rutherford, 2006*; *Pierce & Cowling, 1984*). Both fynbos and thicket shrublands are evergreen. Fynbos shrublands are fire-prone and flammable while smaller areas of thicket vegetation seldom burn (*Kraaij & van Wilgen, 2014*). The persistence of fynbos-thicket mosaics requires fire at appropriate intervals (15–25 years) since thicket may become dominant in the prolonged absence of fire (*Kraaij & van Wilgen, 2014*; *Strydom et al., 2020*, *2021*). The region harbours extensive infestations of woody IAPs, commonly of the genera *Acacia*, *Eucalyptus*, and *Pinus* (*Baard & Kraaij, 2014*; *van Wilgen et al., 2016*), which are deemed to alter the occurrence and behaviour of fires (*Kraaij, Cowling & van Wilgen, 2011*; *Kraaij et al., 2018*).

### Data collection

Flammability was assessed in relation to the fuel traits of 30 shrub species (Table 1) – ten from each of the three vegetation groups: IAPs, fynbos, and thicket. All alien and native

**Table 1** Study species from three vegetation groups (invasive alien plants, fynbos, and thicket) for which flammability was assessed in relation to fuel traits.

| Invasive alien plants | Fynbos | Thicket |
|---|---|---|
| Fabaceae | Asteraceae | Anacardiaceae |
| *Acacia cyclops* G.Don | *Metalasia muricata* (L.) D.Don | *Searsia lucida* (L.) F.A.Barkley |
| *Acacia mearnsii* De Wild. | *Passerina rigida* Wikstr. | Asteraceae |
| *Acacia melanoxylon* R.Br. | Ericaceae | *Tarchonanthus littoralis* P.P.J.Herman |
| *Acacia saligna* (Labill.) Wendl | *Erica canaliculata* Andrews | *Osteospermum moniliferum* L. |
| Myrtaceae | *Erica discolor* Andrews | Ebenaceae |
| *Callistemon viminalis* (Sol. ex Gaertn.) G.Don | Fabaceae | *Diospyros dichrophylla* (Gand.) De Winter |
| *Eucalyptus camaldulensis* Dehnh. | *Aspalathus spinosa* L. | Celastraceae |
| *Leptospermum laevigatum* (Gaertn.) F.Muell. | Proteaceae | *Cassine peragua* L. |
| Pinaceae | *Leucadendron eucalyptifolium* H. Buek ex Meisn. | *Gymnosporia buxifolia* (L.) Szyszyl. |
| *Pinus pinaster* Aiton | Rhamnaceae | *Pterocelastrus tricuspidatus* Walp. |
| *Pinus radiata* D.Don | *Phylica axillaris* Lam. | Salicaceae |
| Solanacaeae | Rubiaceae | *Scolopia zeyheri* (Nees) Szyszyl. |
| *Cestrum laevigatum* Schltdl. | *Cliffortia ericifolia* E.Mey. ex Harv | Santalaceae |
| | *Cliffortia ilicifolia* L. | *Osyris compressa* A.DC. |
| | Rutaceae | Sapotaceae |
| | *Agathosma ovata* (Thunb.) Pillans | *Sideroxylon inerme* L. |

**Note:**
Plant families are indicated, and nomenclature follows *The Plant List (2013)*.

study species commonly occur in the coastal fynbos and thicket of the study region (*Baard & Kraaij, 2014*; *Rebelo et al., 2006*; *Strydom et al., 2020*) and were chosen for ease of material collection and to be representative of flammability of the vegetation of the region. Flammability of plant shoots (hereafter samples) of the different species was experimentally measured using the method and equipment described by *Jaureguiberry, Bertone & Diaz (2011)*. The apparatus comprises a metal barrel (85 cm × 60 cm) that is horizontally orientated with the top removable half that is used for wind protection (*Baeza et al., 2002*). The metal barrel is connected to a grill thermometer, removable gas cylinder and a blowtorch (*Cubino et al., 2018*; *Jaureguiberry, Bertone & Diaz, 2011*). Samples comprised sun-exposed terminal branches that were approximately 70 cm in length that were representative of the fuel structure of the species. As detailed in *Msweli et al. (2020)*, the flammability experiments were conducted on 21 different occasions during February–November 2018 across a range of weather conditions. Samples were kept in closed plastic containers after collection prior to burning, and burning was completed within 4 h of sample collection to minimise moisture loss. Samples were burnt outdoors using an approach similar to that of *Calitz, Potts & Cowling (2015)*. Each sample was placed on the barrel cavity grill to pre-heat at 230 °C for 2 min to imitate the heating and drying effect of an approaching fire. If the samples had not spontaneously ignited within two minutes, it was ignited at the top of the shoot by exposing it to the blow torch for a period of five seconds (*Calitz, Potts & Cowling, 2015*). Flammability measures recorded were (i) combustibility, taken as the maximum temperature reached by the burning

sample, measured using an infrared thermometer (Major Tech 695; maximum recordable temperature: 800 °C) after *Jaureguiberry, Bertone & Diaz (2011)* and *Cui et al. (2020)*; (ii) completeness of burn (consumability), calculated as the proportion of the pre-burn wet mass of the samples that was consumed by the fire; and (iii) time-to-ignition (ignitability), measured as the time elapsed between placement of the samples on the grill and spontaneous ignition (appearance of the first flame). Samples that did not spontaneously ignite within 120 s of pre-heating were ignited with a blow torch and assigned an arbitrary time-to-ignition of 200 s (to be clearly differentiated from 120 s, but still depictable on graph scales).

The wet mass of samples was recorded prior to conducting the flammability experiments. On each of the 21 occasions that the flammability experiments were performed, a duplicate set of plant samples were collected and oven-dried at 80 °C for 48 h and reweighed to obtain dry fuel mass (*Ruffault et al., 2018*). Live fuel moisture content was calculated as the percentage of wet mass that comprised water. The dry biomass of samples was regarded to be a proxy for the fuel loading that samples presented. Seeing that those samples subjected to flammability experiments could not be dried beforehand, the dry biomass of burnt samples was estimated from their wet biomass prior to being burnt, and the live fuel moisture content measured for the duplicate set of plant samples (where dry biomass = pre-burn wet biomass – (pre-burn wet biomass × proportion of fuel moisture)). For the flammability response variables, and for the fuel traits, namely live moisture and dry biomass, we thus had 21 replicate values per species.

Other fuel traits of interest were the proportion of fine fuels, coarse fuels, dead fuels, and fuel bed porosity. To measure these fuel structural traits, a one-off collection of six samples per species was conducted, similar to those collected for the flammability experiments and following the approach by *Burger & Bond (2015)*. Three samples were used to measure the mass of live material in different fuel size classes. Each of these samples was separated based on stem diameter into fine fuels (<3 mm) and coarse fuels (>6 mm) (in the interest of brevity, we disregarded 3–6 mm fuels as results pertaining to this category mirrored those of the fine fuels). Leaves were included in the stem diameter class to which they were attached and the plant mass in each size class was weighed. The samples were also separated into live and dead fuel material (twigs, branches, and leaves) and subsequently weighed (respectively). The remaining three samples of each species were used to determine fuel bed porosity, calculated as the canopy volume (based on the formula for the volume of a cone, as this geometrical shape best approximated the shape of our shoot samples) divided by the fuel volume (after *Burger & Bond, 2015*). The latter was the volume occupied by the samples and measured through means of volume displacement in a 5 L measuring bucket.

## Data analysis

The response variables (maximum temperature, completeness of burn, and time-to-ignition) and fuel traits, namely live fuel moisture and dry biomass, were derived from the flammability experiments using averages of the 21 replicates for each of the 30 sampled species. The response variables did not violate assumptions of normality according to

**Table 2 Multiple regression model results for flammability (maximum temperature, completeness of burn, and time-to-ignition, respectively) in relation to fuel traits as fixed factors, *i.e.*, proportion of fine fuels, coarse fuels, dry biomass, and live fuel moisture.**

| Fixed factors | Fine fuels | | Coarse fuels | | Dry biomass | | Fuel moisture | | Model statistics | |
|---|---|---|---|---|---|---|---|---|---|---|
| | t[a] | Scaled est.[b] | t[a] | Scaled est.[b] | t[a] | Scaled est.[b] | t[a] | Scaled est.[b] | F[a] | R² adj.[a] |
| Maximum temperature | | | −1.7 | −0.32 | 3.8*** | 0.68 | −2.4* | −0.36 | 11.87 | 0.53 |
| Completeness of burn | 2.2* | 0.45 | | | 1.9 | 0.37 | −1.5 | −0.27 | 4.54 | 0.27 |
| Time-to-ignition | −1.9 | −0.33 | | | | | 1.7 | | 4.87 | 0.21 |

Notes:
Results shown are for the preferred models after stepwise selection (details in Supplemental S2).
Significance codes: *$p < 0.05$, ***$p < 0.001$.
[a] t statistic, F statistic, and R² adjusted (adj.) obtained from the multiple regression model output.
[b] Scaled estimates were derived from incorporating the scale function in the multiple regression model.

Shapiro–Wilk test (maximum temperature: W = 0.97, $p$ = 0.66; completeness of burn: W = 0.96, $p$ = 0.38; and time-to-ignition: W = 0.96, $p$ = 0.27). For the other fuel traits, namely the proportion of fine fuels, coarse fuels, and dead fuels, and fuel bed porosity (ratio), we used averages of the three replicates measured per species. A combined dataset was created for further analyses containing, for all the variables, the averages per species, with the 30 species thus comprising unique data points.

All statistical analyses were performed in the open-source R software version 3.6.1 (*R Development Core Team, 2019*). Multiple regression models fitted with the lm() function (*Chambers, 1992*) were used to assess the relationships between flammability responses (respectively) and the following fuel traits as fixed (explanatory) factors: (i) proportion of fine fuels, (ii) proportion of coarse fuels, (iii) proportion of dead fuels, (iv) dry biomass, (v) fuel bed porosity (ratio), and (vi) live fuel moisture (percentage). Stepwise model selection based on the lowest Akaike information criterion (AIC) (*Sakamoto, Ishiguro & Kitagawa, 1986*) was used to choose the best combination of fixed factors that could potentially predict flammability responses (respectively), but results of the preferred models were compared with those of the full models given potential bias associated with stepwise selection procedures (*Smith, 2018*). The scale function (*Becker, Chambers & Wilks, 2018*; *Hastie & Pregibon, 1992*) was incorporated to the multiple regression models to standardize variables of different scales and obtain the relative influence of each fixed factor. To test if the flammability responses and fuel traits differed among vegetation groups (IAPs, fynbos, and thicket), we employed Kruskal–Wallis *H* test (as most of the fuel trait variables did not conform to normality) and thereafter Dunn's test for multiple comparisons if significant differences occurred (*Dunn, 1964*).

## RESULTS

### Effects of fuel traits

The stepwise selection procedure retained different combinations of fixed factors for the respective flammability responses, but the proportion of dead fuels and fuel bed porosity were not retained by any of the preferred models (Table 2; detailed outputs in Supplemental S2). Live fuel moisture was retained in the preferred models for all the respective flammability measures, and lower live fuel moisture significantly increased

maximum temperature (Table 2, Fig. 1). No significant relationships were found between live fuel moisture and completeness of burn; live fuel moisture and time-to-ignition; and fine fuels and time-to-ignition. Greater dry biomass increased combustibility by significantly increasing maximum temperature. Amongst the assessed fixed factors, dry biomass had the largest influence (*i.e.*, the largest scaled estimates; Table 2) on maximum temperature. Fine fuels significantly increased completeness of burn in the preferred model, although it was not significant in the full model, where dry biomass, instead, enhanced completeness of burn (Supplemental S3). Bar this discrepancy, the results from the preferred models concurred with those of the full models (Supplemental S3).

## Vegetation group comparisons

Maximum temperature and live fuel moisture did not differ significantly among the vegetation groups, whereas completeness of burn, time-to-ignition, proportion of fine fuels, proportion of coarse fuels, fuel bed porosity, and dry biomass differed significantly (Fig. 2; details in Supplemental S4). Completeness of burn did not differ between species of IAPs and fynbos but was significantly higher in these vegetation groups than in thicket species. Time-to-ignition did not differ between species of IAPs and fynbos but was significantly shorter in these vegetation groups than in thicket species. Fynbos species had a significantly higher proportion of fine fuels than IAP species and thicket species, whereas thicket species had a significantly higher proportion of coarse fuels than species of IAPs and fynbos. Fuel bed porosity was significantly higher in fynbos species than in thicket species, while IAP species did not differ from species of fynbos or thicket. Dry biomass was significantly lower in fynbos species than in thicket species, while IAP species did not differ from the other vegetation groups.

## DISCUSSION

### Relation between fuel traits and flammability

We assessed how the fuel traits of 30 woody shrub species affected their flammability. This assessment was more comprehensive than most other studies of this nature (compare Supplemental S1) in terms of the diversity of flammability measures and fuel traits assessed, and the wide range of species from different vegetation types considered. Our results confirmed a positive relationship between dry biomass (fuel loading) and combustibility (maximum temperature). The enhancing effect of the amount of biomass (fuel loading) that vegetation presents on combustibility, fire intensity or burn severity was observed in several other studies (*Keeley, 2009*; *Saura-Mas et al., 2010*; *Simpson et al., 2016*) at the scale of the individual plant (or plant components) and at vegetation community scale. Accordingly, a positive relationship between fuel load and burn intensity was also observed in Australian forests and woodlands, Californian shrublands, and South African ecosystems (*Kraaij et al., 2018*; *Schwilk, 2003*; *Simpson et al., 2016*).

The proportion of fine fuels in our study enhanced completeness of burn and was retained by the preferred model for time-to-ignition, exhibiting a weak positive relationship with ignitability. However, the proportion of fine fuels did not influence maximum temperature. Other flammability experiments conducted at the scale of plant

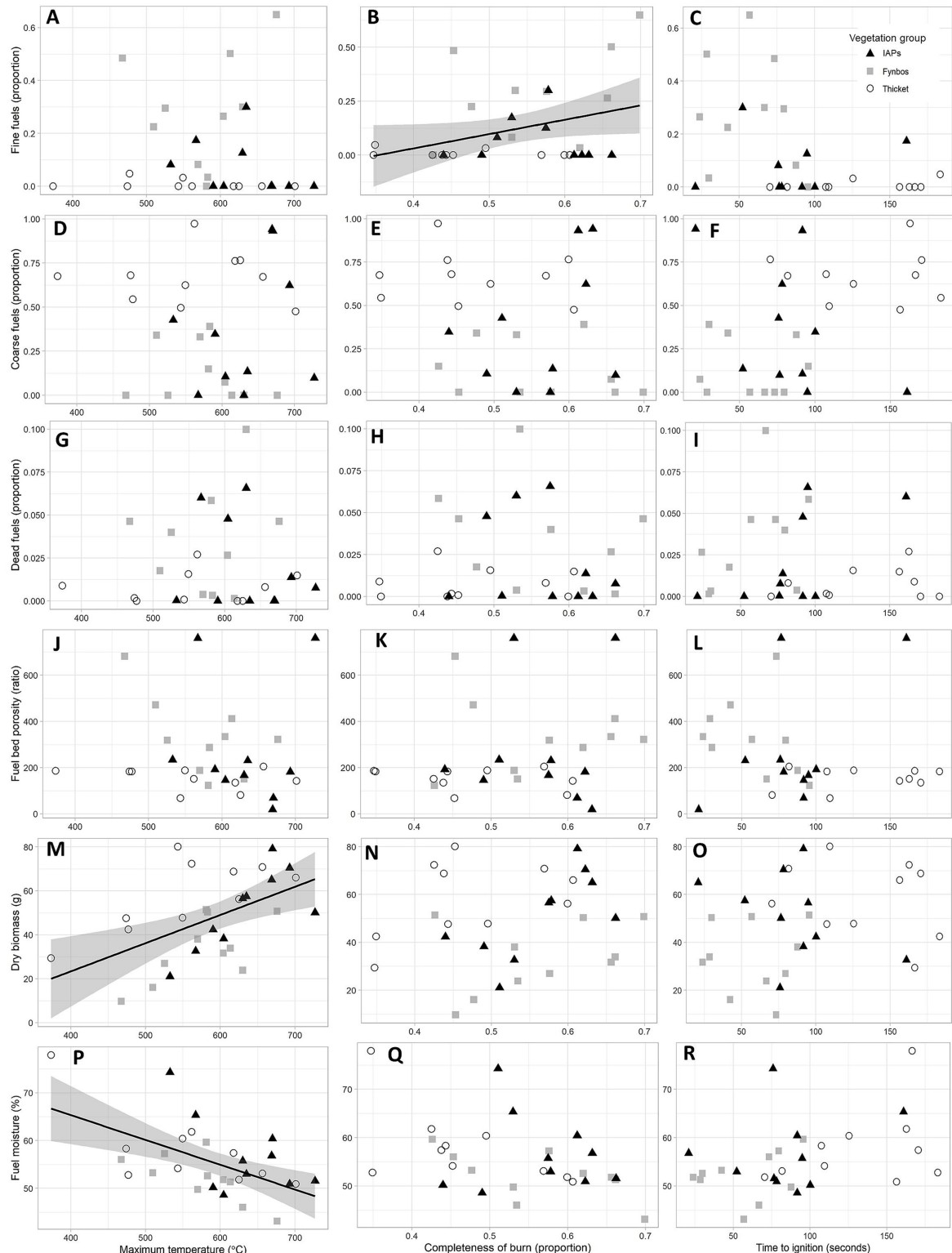

**Figure 1 Relationships between flammability measures (maximum temperature, completeness of burn, and time-to-ignition), and fuel traits: (A–C) fine fuels, (D–F) coarse fuels, (G–I) dead fuels, (J–L) porosity, (M–O) dry biomass, and (P–R) fuel moisture.** Each point represents the average value for a species, and points are formatted based on vegetation group association. Lines and shaded confidence interval bands are indicated for relationships shown to be statistically significant in preferred multiple regression models (details in Table 1 and Supplemental S2).

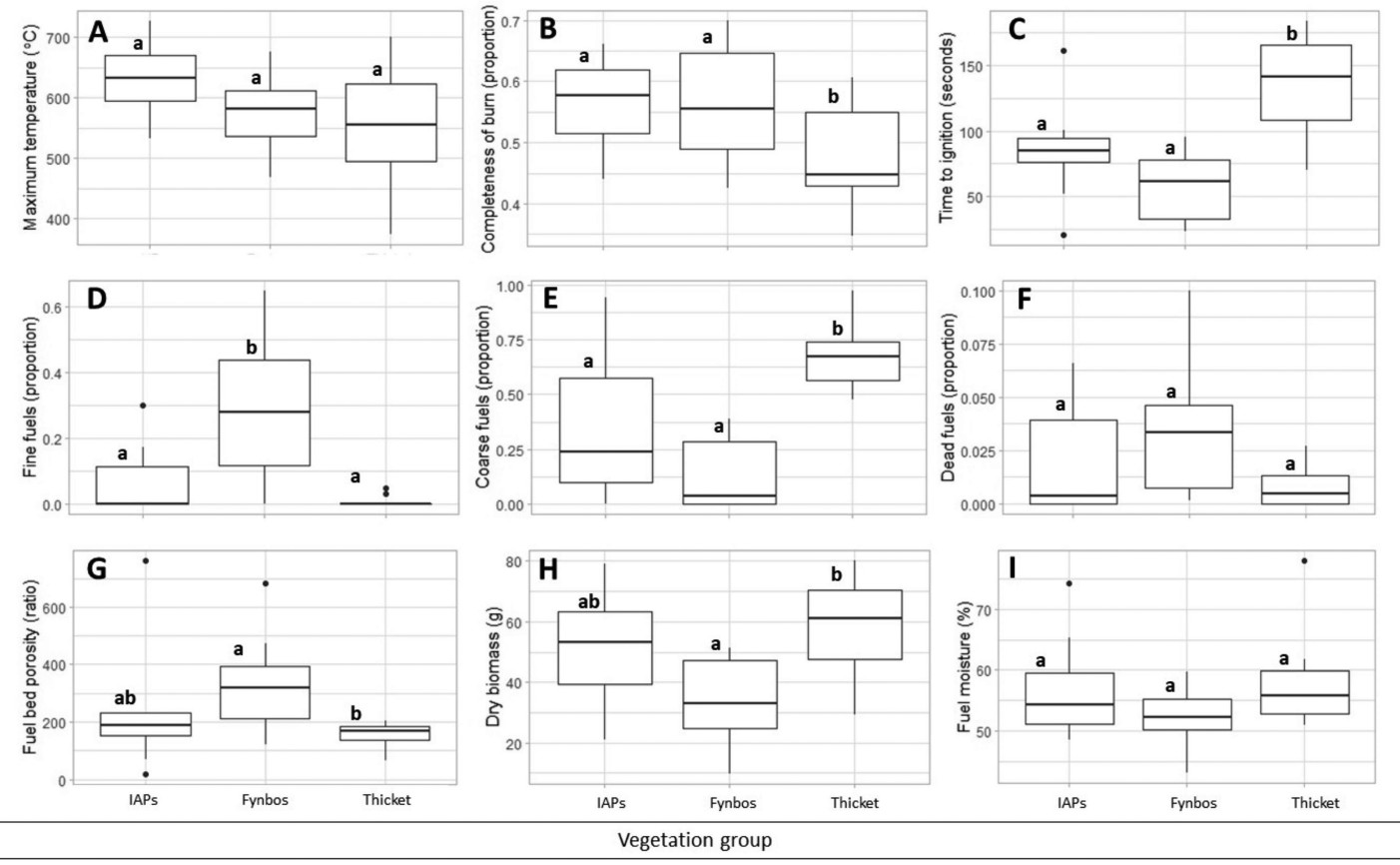

**Figure 2 (A–C) Flammability measures and (D–I) fuel traits compared among vegetation groups (IAPs, invasive alien plants; fynbos; thicket).** Boxes show 25–75 quantile ranges, middle lines show medians, dots show outliers and whiskers show 1.5*interquartile ranges. Disparate small letters denote significant differences among vegetation groups based on Kruskal Wallis *H* test results (details in Supplemental S4) and Dunn's multiple comparisons.

shoots or plots in various vegetation types unanimously showed that fine fuels enhanced flammability (*Burger & Bond, 2015*; *Calitz, Potts & Cowling, 2015*; *Santana & Marrs, 2014*; *Schwilk, 2003*). *Burger & Bond (2015)* found that the proportion of fine fuels and of dead fuels were the most important factors governing completeness of burn in fynbos and forest species, while dead fuel retention also enhanced completeness of burn in other systems (*Davies & Legg, 2011*; *Santana & Marrs, 2014*). In contrast to the findings of *Burger & Bond (2015)*, we found that the proportion of dead fuels did not affect any of the flammability measures. This may be due to the sampling of sun-exposed (younger) branch tips that had very little dead material (the maximum proportion of dead fuels in our study was 0.10 *vs.* a mean for flammable species of 0.15 and a maximum of 0.41 reported by *Burger & Bond, (2015)*). In a study of the flammability of Mediterranean basin shrubs, *Pellizzaro et al. (2007)* accordingly noted an absence of dead fuels on terminal branches. The other factor that had no effect on flammability in our study was fuel bed porosity. Likewise, fuel porosity had inconsistent effects among studies and on the different components of flammability (Supplemental S1).

The impeding effects of fuel moisture on flammability and fire behaviour are widely recognized in different ecosystems such as shrublands and forests (*Alam et al., 2020*; *Alessio et al., 2008*; *Bianchi & Defossé, 2015*; *Davies & Legg, 2011*; *Dimitrakopoulos, 2001*; *Pausas et al., 2004*; *Santacruz-García et al., 2019*; *Saura-Mas et al., 2010*), although grasses, despite high moisture contents, may ignite readily (*Cubino et al., 2018*) on account of small leaf surface area that allows quick moisture evaporation to enable fuel ignition (*Simpson et al., 2016*). In our study, live fuel moisture significantly reduced combustibility but did not significantly affect completeness of burn and ignitability. Although live fuel moisture was retained by the preferred models for all the flammability measures, the magnitude of its effects on flammability was relatively low (small scaled estimates, Table 2). Some other studies show strong relationships between fuel moisture and flammability measures (*i.e.*, rate of spread and ignitability) (*Alessio et al., 2008*; *Davies & Legg, 2011*; *Dimitrakopoulos, 2001*; *Saura-Mas et al., 2010*; *Simpson et al., 2016*), but also non-significant effects of fuel moisture on flammability (*Burger & Bond, 2015*; *Cubino et al., 2018*), or inconsistent relations between fuel moisture and ignitability (*Fletcher et al., 2007*). Our study did not show any overriding effects of live fuel moisture on flammability.

Generally, discrepancies in the scale of experimentation and the methods and measures used severely complicated comparisons of the effects of fuel traits on flammability. For instance, flammability of leaves may not resemble flammability of plant shoots (*Alam et al., 2020*); flammability of leaves or shoots may not resemble whole plant flammability (*Wyse et al., 2016*), which in turn may not resemble monospecific stand-level flammability (*Schwilk, 2003*; *Santana & Marrs, 2014*); and ultimately, extrapolation from species-level flammability to community-level (species-mixes) flammability is complex (*Cubino et al., 2018*; *Wyse, Perry & Curran, 2018*). The study of flammability also requires discernment between canopy fires largely in live fuels, and surface fires in cured grass swards or leave litter beds and thus largely dead fuels (*Murray, Hardstaff & Phillips, 2013*; *Simpson et al., 2016*), or combinations and thus transmission of fire between different fuel layers (*Burton et al., 2021*; *Santana & Marrs, 2014*). The methods used to achieve ignition in flammability studies (whether fuels are dried beforehand, *e.g.*, compare *Burton et al., 2021*; *Cubino et al., 2018*; *Murray, Hardstaff & Phillips, 2013*; *Wyse et al., 2016*) furthermore has relevance for meaningful interpretation of results. Drying of fuels prior to flammability assessment would, for instance, be ill-suited to the investigation of flammability of evergreen shrublands that sustain canopy fires (such as the study system). Hence, care needs to be taken with extrapolation of flammability-trait relationships observed in limited experiments to community-scale fire behaviour (*Cubino et al., 2018*), and with comparison of flammability studies generally (such as our attempt in Supplemental S1).

## Vegetation group comparisons

In line with the results of our earlier study that explored flammability in relation to fire weather (*Msweli et al., 2020*), the averaged vegetation comparisons showed that species of fynbos and IAPs exhibited greater flammability on account of higher completeness of burn and more rapid ignition than thicket species. There was no apparent distinction between fynbos species and IAP species in terms of their flammability, although there was a weak

trend of IAPs burning at higher intensities than fynbos and thicket species. This aligns with an investigation conducted at vegetation community scale which showed that burn severity (deduced from the differenced Normalised Burn Ratio) in the extreme 2017 fires in the region was higher in vegetation dominated by invasive alien plants than in native thicket and fynbos (*Kraaij et al., 2018*). In the current study, species of fynbos and IAPs did not differ in terms of most fuel traits investigated, except for the proportion of fine fuels. In accordance with other studies that compared species from various South African vegetation types (*Burger & Bond, 2015*; *Calitz, Potts & Cowling, 2015*; *van Wilgen, Higgins & Bellstedt, 1990*), fynbos species in our study had large proportions of fine fuels, high porosity and low fuel loadings, which likely accounted for high flammability, and in particular, ignitability. In addition, *Burger & Bond (2015)*, showed that flammability of fynbos species was enhanced by a large proportion of dead fuels. Although not statistically significant in our study, the proportion of dead fuel was somewhat higher in fynbos species than in IAPs and thicket species.

IAP species displayed relatively high combustibility and consumability which could not be clearly linked to the fuel traits examined. For instance, IAP species showed rapid ignition comparable to that of fynbos species, despite their relatively high fuel loadings approximating those of thicket species. Generally, IAP species presented a combination of fire-prone (*i.e.*, high fuel loading) and fire-resistant (*i.e.*, low proportion of fine fuel and low porosity) fuel traits which suggested that other fuel traits not accounted for, such as volatile organic compounds (*Dimitrakopoulos, 2001*; *Saura-Mas et al., 2010*), likely increased the flammability of IAPs. The 10 sample species of IAPs furthermore reflected a random set of species with diverse origins rather than a community that evolved collectively under a particular fire regime, which likely introduced variability in fuel traits and flammability response. Accordingly, extreme outliers were evident in some flammability measures, such as time to ignition (Fig. 2C), where *Pinus radiata* and *Acacia saligna* displayed exceedingly rapid and slow ignition, respectively. These results prompt for a more detailed investigation of flammability of IAPs in relation to their fuel traits at a species-specific level and consideration of fuel chemical contents that may enhance flammability (*Burton et al., 2021*; *Santacruz-García et al., 2019*).

Thicket species had high proportions of coarse and dense fuels, which accounted for this vegetation's low flammability, as was also previously observed (*Burger & Bond, 2015*; *Calitz, Potts & Cowling, 2015*; *Pierce & Cowling, 1984*). Although continuous (~dense) fuels generally facilitate fire spread (*Keeley, 2009*), high fuel density and coarse fuels can limit oxygen supply to the fire and therefore reduce the rate of fire spread (*Scarff & Westoby, 2006*). In our study, thicket species had high fuel loadings but low completeness of burn, corresponding with the earlier study of the 2017 fires in the region which indicated high fuel biomass, but small areas burnt, of thicket compared to fynbos vegetation (*Kraaij et al., 2018*).

Live fuel moisture was indistinguishable among vegetation groups. The lack of difference between the live fuel moisture contents of fynbos and thicket species, in particular, was contrary to expectation, given that *van Wilgen, Higgins & Bellstedt (1990)* found foliar moisture content to be 50–100% higher in forest trees (which share many

species with thicket; *Strydom et al., 2021*) than in fynbos. *Msweli et al. (2020)* found that, compared to live fuel moisture, fire weather had more significant effects on flammability, but that live fuel moisture did not correlate with fire weather conditions. Based on these findings, *Msweli et al. (2020)* argued that the importance of live fuel moisture for flammability of evergreen shrublands likely rests on inter-specific and inter-vegetation type differences in fuel moisture contents. However, here we found no significant differences between the fuel moisture contents of common species from the assessed vegetation types. Likewise, flammability differences between seeding and non-seeding species in the Mediterranean Basin could not be attributed to differences in fuel moisture, and chemical content was invoked as a potential contributing factor (*Saura-Mas et al., 2010*). Live fuel moisture still warrants further investigation at species-level, but does not appear to primarily account for differences in flammability among the vegetation groups considered in our study.

## CONCLUSIONS

This study of flammability in relation to fuel traits of a diversity of native and alien invasive shrub species showed that increases in fuel loading and reductions in live fuel moisture enhanced combustibility, whereas increases in fine fuels enhanced consumability. Fuel bed porosity and the proportion of dead fuels had little effect on flammability. Little distinction in flammability was evident between species of IAPs and fynbos, but both these vegetation groups were significantly more ignitable than thicket species. Fuel traits most notably associated with particular vegetation types included large proportions of fine fuels in fynbos species, and high fuel loading and coarse fuels in thicket species. Surprisingly, live fuel moisture did not differ among the vegetation groups and did not have major effects on flammability. Detailed species-level investigation of flammability in relation to fuel traits, including fuel chemical composition, is suggested to inform the fire risk posed by particular IAP species relative to native vegetation. Such assessments will have relevance for future management of fire regimes.

## ACKNOWLEDGEMENTS

We thank Nicole Malan and Natasja Van Zyl for assistance with specimen collection and measurements. We thank Curtis Daehler, Kevin Faccenda, Owen Price and an anonymous reviewer for constructive criticism which led to improvement of the manuscript.

### Funding

Funding for this study was provided by the African Centre for Coastal Palaeoscience, Postgraduate research scholarship from the Nelson Mandela University, and German Academic Exchange Service – National Research Foundation Masters scholarship. The funders had no role in study design, data collection and analysis, decision to publish, or preparation of the manuscript.

## Grant Disclosures

The following grant information was disclosed by the authors:
African Centre for Coastal Palaeoscience.
Nelson Mandela University.
German Academic Exchange Service – National Research Foundation Masters scholarship.

## Competing Interests

Alastair J. Potts is an Academic Editor for PeerJ. The authors have no other competing interests to declare.

## Author Contributions

- Tineke Kraaij conceived and designed the experiments, analyzed the data, prepared figures and/or tables, authored or reviewed drafts of the article, and approved the final draft.
- Samukelisiwe T. Msweli conceived and designed the experiments, performed the experiments, analyzed the data, prepared figures and/or tables, authored or reviewed drafts of the article, and approved the final draft.
- Alastair J. Potts conceived and designed the experiments, analyzed the data, authored or reviewed drafts of the article, and approved the final draft.

## Data Availability

The raw data and the R codes are available in the Supplemental Files.

## Supplemental Information

Supplemental information for this article can be found online at http://dx.doi.org/10.7717/peerj.13765#supplemental-information.

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
