# Peer review of "Fuel trait effects on flammability of native and invasive alien shrubs in coastal fynbos and thicket (Cape Floristic Region)"

_PeerJ, doi:10.7717/peerj.13765_

## Round 0.1 · original submission · Major Revisions

The reviewers have provided various comments and considerations that should be addressed in a revised manuscript. Please pay particular attention to concerns raised regarding methodology, terminology, data reporting and analyses.

·

Basic reporting

In both the introduction and abstract, you talk about the 2017 fires but then never mention them again in the discussion. The discussion should either touch on them again, or remove mention of them from the paper. Conceptually it seems like they are the inspiration of the paper, but do not play any sort of critical part in the research.

Throughout the abstract and paper, rather than saying that [habitat] had [fire trait], say species from [habitat] had [fire trait] e.g. “[species from] Fynbos and IAPs generally exhibited greater flammability” (line 31) as you assessed everything at the species, rather than ecosystem level. You also only sampled 10 species from each habitat type and generalizing to all species from that community does not seem to be supported by the data.

Supplementary 1 can be expanded as it is missing several recent experimental studies of plant flammability at the branch level. References to the following may also be appropriate in the main text. References cited in these papers also include more recent ecosystem level studies which are likely also relevant.

Alam MA, Wyse SV, Buckley HL, Perry GL, Sullivan JJ, Mason NW, Buxton R, Richardson SJ, Curran TJ (2020) Shoot flammability is decoupled from leaf flammability, but controlled by leaf functional traits. J Ecol 108:641–653. https://doi.org/10.1111/1365-2745.13289

Burton JE, Cawson JG, Filkov AI, Penman TD (2020) Leaf traits predict global patterns in the structure and flammability of forest litter beds. J of Ecol 109:1344–1355. https://doi.org/10.1111/1365-2745.13561

Cornwell WK, Elvira A, van Kempen L, van Logtestijn RS, Aptroot A, Cornelissen JHC (2015) Flammability across the gymnosperm phylogeny: the importance of litter particle size. New Phytol 206:672–681. https://doi.org/10.1111/nph.13317

Cui X, Paterson AM, Wyse SV, Alam MA, Maurin KJ, Pieper R, Cubino JP, O’Connell DM, Donkers D, Bréda J, Buckley HL (2020) Shoot flammability of vascular plants is phylogenetically conserved and related to habitat fire-proneness and growth form. Nat Plants 6:355–359. https://doi.org/10.1038/s41477-020-0635-1

Santacruz-García AC, Bravo S, del Corro F, Ojeda F (2019) A comparative assessment of plant flammability through a functional approach: the case of woody species from Argentine Chaco region. Austral Ecol 44:1416–1429. https://doi.org/10.1111/aec.12815

Wyse SV, Perry GL, O’Connell DM, Holland PS, Wright MJ, Hosted CL, Whitelock SL, Geary IJ, Maurin KJ, Curran TJ (2016) A quantitative assessment of shoot flammability for 60 tree and shrub species supports rankings based on expert opinion. Int J Wildland Fire 25:466–477. https://doi.org/10.1071/WF15047

Wyse SV, Perry GL, Curran TJ (2018) Shoot-level flammability of species mixtures is driven by the most flammable species: implications for vegetation-fire feedback favouring invasive species. Ecosystems 2:886–900. https://doi.org/10.1007/s10021-017-0195-z

Supplementary 2 should be made as table 1 as the selection of species will be of interest to many readers and is important to make the species identities clear considering only 10 species were sampled per vegetation group. It would also be of value to add the family for each species and sort it by the family within the vegetation groups. It is unclear how the species are currently ordered. “Gymnosporia buxifolia (L.)” and “Scolopia zeyheri (Nees)” are not correct names per ICN as they are missing the revising author.

The description of Table 1 appears cut off at the end.

Table 1 was also not intuitive to me and I had to look at it side-by side with supp. 3 to make sense of it. I suggest switching the rows and columns as it is derived from a glm which is typically written from left to right.

In figure 1, you have enough space to include the entire word “thicket” and “fynbos” and I suggest you use that instead of the abbreviation. Same in the legend of Figure 2.

In figure 1 I suggest changing the colors to those which are better for those who are red/green colorblind. Making the points different colors and shapes would also be beneficial in case this ever gets printed as black and white copies.

In figure 2 you also have enough room to not abbreviate vegetation group.

In your discussion, it would be useful to break up your 2nd paragraph into two or more, one comparing your results to studies which assessed flammability at the shoot level like your study, and others that assessed it at a plot or ecosystem level. While shoot level flammability is generally correlated with ecosystem level of flammability, there are differences and opportunities for discussion about those differences (like you have already done for dead material). Incorporation of the citations given above would also be beneficial.

Hypotheses are clear and the data test them directly.

Experimental design

Line 143: Please elaborate in the text how the 30 species were chosen. What were the criteria you used for selection?

Line 155: “samples that did not spontaneously ignite within 120 seconds of pre-heating were ignited with a blow torch and assigned an arbitrary time-to-ignition of 200 seconds.” Why did you use this number specifically?

Line 162: “The dry mass of samples was regarded to be a proxy for the fuel load that samples presented.” How do you justify this? Under most definitions fuel load is the quantity of fuels contributed by a species divided by a certain area of habitat, therefore rare species would contribute a smaller fuel load than common species. Your definition of this entirely ignores the area component. Methodologically, this is the weakest part of the paper and unless you have a very good justification, I suggest you reanalyze these data but either recalculate fuel load with different data, rename it as “sample biomass”, or omit it entirely from the analysis.

Line 174: “ Each of these samples was separated based on stem diameter into fine fuels (< 3 mm) and coarse fuels (> 6 mm).” Why did you not consider 3-6 mm stems? Burger & Bond consider them as a separate category.

Line 178: “calculated as the canopy volume (based on the formula for the volume of a cone)” How do you justify this? Did you specifically choose samples which were approximately conical? Can you add clarification to the methods? I could see an argument for a cylinder (or even a rectangular prism) being a better representation of the volume depending on the sample.

Line 189: “A combined dataset was created for further analyses containing, for all the variables, the averages per species, with the 30 species thus comprising replicates.” I don’t think replicate is the right word to use here since each species is different. Maybe replace replicates with “unique data points”?

Line 193: “Multiple regression models were used” Can you clarify what function you used in the text? lm()?

For the analysis shown in figure 2, did you consider using the unaveraged results between each species (e.g. in a-c analyzing the 21 replicates for each species). As it currently stands, your data is taking an average of an average which affects your error bars and significance tests. For analyzing the unaveraged data you could try using a glmm or gam (depending on normality) with species as a random factor and habitat as the fixed factor.

Validity of the findings

Data are provided in the supplementary material. Providing the R code would also be beneficial.

All analyses and discussion make sense and are supported by the data with the exception of fuel load per my above comments.

Conclusions are clear and supported by the data.

Additional comments

Overall, this is a well written paper with good data and analysis. More recent research in this area needs to be reviewed and the citations above should hopefully be helpful in that regard.

The main reason I suggest major revisions is due to the fuel load issue.

Reviewer 2 ·

Basic reporting

This manuscript presents data from flammability trials on 30 shrub species from the southern Cape of South Africa. The authors sampled species from two vegetation types and also included a set of invasive alien species. This area of work is relatively undeveloped, therefore I think there is scope for exploratory analyses such as these. I did find that the introduction and framing did not provide much explanation or justification for the study. For example, is there field-based evidence that these vegetation types have different fire behavior? From my own knowledge of fynbos and thicket, I would expect higher flammability among typical fynbos species (but with plenty of exceptions). The authors, however, never lay out any background on this.

This comparison across vegetation types is one of two main goals of the manuscript and should be better justified. I also found the category of invasives confusing. Is there theory predicting that non native invasive species should have a particular flammability? The second purpose of the manuscript was to determine important traits that predict flammability. But I found the justification here also a bit confusing. Why were those particular traits measured? The paragraph that starts on line 55 seems to be setting up a justification but it is muddled and ends with a rather weak one: "fuel bed porosity and proportion of fine and dead fuels have received less attention". Perhaps with good reason? Is there any proposed link to the plant ecology that would justify that attention? Or is the goal to find something easier to measure that will predict flammability without the need for trials? What is the actual goal of the trait portion of the study?

There are some references in the text that are missing from the literature cited.

Supplementary table 1 is a good start but is woefully incomplete. Why that particular selection of studies? There are many other important studies omitted. Off hand I can think of several from New Zealand, several from North America, and several missed European studies. The missed literature includes studies on leaf litter flammability in forests, and others on canopy flammability across shrubs and grasses. I also do not agree with some of the characterization of "fuels traits" as listed in that table. The authors don't distinguish canopy from litter flammability throughout despite these fuel types involving very different vegetation-fire feedbacks.

Experimental design

My main concern was that the flammability trials were not well described. I had trouble figuring out how exactly the authors measured the main flammability parameter reported ("burn intensity"). On line 151, the authors cite Keeley 2009 implying that the temperature measure used follows from that citation. Keeley (2009), however does not define the term "burn intensity" but rather discusses uses of the term "fire intensity". However, even broad definitions of fire intensity include the idea of rate of heat release/transfer which is not the same as a peak temperature measured on an unknown material. What do the authors mean by "the maximum temperature reached by a sample during burning"? Temperature of what? gas temperature? Temperature of some object of known thermal properties? I realize that heat transfer is difficult to estimate in flammability trials and I am sympathetic to using proxy measures, but the reader needs to know what exactly is being measured and the justification. Was this number taken from a time series from thermocouples? The original Jaureguiberry et al (2011) paper describes an IR gun aimed at the grill? I'm not sure what that is measuring. I am ok with well-described proxies and have found that this area is an example of where some things work better in practice than in theory: for example, temperature time series summaries from thermocouples often correlate well with other measures of fire behavior. However, the authors should demonstrate that their temperature proxy captures something real about fire behavior. If the authors measured the maximum temperature reached by some object, then I am willing to consider that as a proxy for rate of heat transfer/fire intensity. The figures should be labeled accurately, however, as "peak temperature". I was unconvinced that the "burn intensity" measure was relevant. Is there any field truthing of this measurement? Also note that Jaureguiberry et al (2011) is missing from the literature cited.

How were the samples treated prior to flammability trials? Cut and bagged on ice? Any dry down period prior to burning? This is all important and the various labs using similar plant "barbecues" all use slight variations.

For the comparisons across species groups, it seems here that the authors used species means. That is acceptable but I would have expected a mixed model with species as a random effect --- I have no problem with the approach but was curious.

Validity of the findings

This paper includes an a priori test of flammability differences across three species groups (fynbos, thicket, and invasives). That is fine, although I would like to see some explanation of how species were chosen and how sampling was randomized. I'm not sure why "invasives" would be thought to have a particular flammability. Perhaps the authors can discuss the types of invasive species (traits, lineages?) found in these communities? I don't believe that the realm of inference for the "IAP" group is really all invasive alien plants and the authors recognize this as they state in the Discussion on lines 304-306! Across the literature, there are examples of invasive species increasing fire spread rates and occurrence (eg grass fire cycle) and invasives decreasing ignitability and spread (some shrub encroachment examples).

I'm not sure what the journal's exact requirements are for data and code. I see that summarized species means are provided, but there is not sufficient data nor any code with which to reproduce the analyses.

Additional comments

In summary: the manuscript would benefit from 1) clearer goals, 2) clearer justification for the exploratory vs the hypothesis testing portions, 3) better description of the flammability trials including some attention to the physics.


Minor comments

- The acronym IAP is not needed. Acronyms hinder understanding and don't really save space.
- This manuscript has both exploratory statistics (the trait-flammability model selection) and hypothesis testing (tests of flammability differences across the species groups). A better explanation of why each approach was used would improve the manuscript. I am also skeptical of any stepwise model selection and there are plenty of good criticisms of stepwise selection in the literature. Why not use the full model? Or represent specific hypotheses with separate models? Or even use model averaging if this is truly just exploratory?
- Line 55: I think this terminology is terrible! Is arrangement of leaves really "extrinsic"? This is an idiosyncratic terminology that is not well accepted.
- Line 73: But some studies explicitly exclude total mass and don't treat fuel load as a "flammability trait" -- presumably because it is so plastic. The framing here cites work from both studies of how leaf litter behaves as fuel and how entire plants (or portions of live plants) behave as fuel. These are different in some important ways but I found the Introduction was not terribly careful about distinguishing them.
- Line 97. Yes, but do you expect some consistent effects or are these likely to be idiosyncratic?
- Lines 257-260. Yes, an issue with treating a single live branch as a random sample of the canopy is that it is likely to misrepresent live vs dead tissue.

·

Basic reporting

Good. There were a few places where more explanation was needed.

Experimental design

Good, except the experimental equipment should have been described

Validity of the findings

No issues

Additional comments

This is a really useful study of flammability and its drivers for a south African region. The paper is well conceived, clearly and concisely written and the conclusions are valid. I have only minor comments, mostly about adding some additional explanation of the methods.
Also, I would prefer if the supplementary information was included as tables in the main text, especially 1 and 2. However, this might be a style issue with the journal.
While I am no expert on flammability studies, my understanding is that the use of different experimental set-ups could have a large influence on results, and therefore this is an important discussion point that was left out.

Specific comments/
L48. Or the leaf scale.
L144. Can you state whether these were the most common shrub species in each vegetation type? It matters because you compare the fuel characteristics of the species among veg types.
L148. You need to briefly explain the experimental set-up, even if it is more fully explained in Msweli 2020. You refer to a grill. What is that?
L299. You need to back this statement up. IAPs were only slightly more intense and complete than fynbos and did not have the shortest time to ignition. So why do you say they were high flammability?

---

## Round 0.2 · Minor Revisions

Two reviewers were satisfied with your responses to previous reviews and you associated revisions to the manuscript. Reviewer #2 has some follow-up comments and suggestions that should be considered. In addition, I have the following minor comments:

L 106 “(Murray et al., 2013), the arrangement...” – should this say “(Murray et al., 2013), or the arrangement...”?

L 216-221 Can you cite a reference for your calculation of porosity? I have seen porosity expressed by others a as a square root value with units as cm (e.g. https://doi.org/10.1016/S0082-0784(73)80092-X) while your approach seems to yield a dimensionless ratio.

I recognize that your Discussion focuses on mean trends across groups and statistically significant (or not) associations with variables. However, I suggest that it would be both interesting and valuable if you could add some brief discussion of individual outlier species. The outlier species may be most important for driving current or future fire regimes. In particular, I wondered about the outlier IAP with shockingly low time to ignition (Fig 2C). Does this species potentially affect risk of ignition for the entire community?

What is most striking to me about the average trends is that IAPs average closer in biomass to the thicket species (e.g. Fig 2H), however in terms of time to ignition the IAPs, on average, behave more like fynbos species with faster ignition (Fig 2C). You might want to consider adding more explicit discussion of this pattern. I would expect that this combination of IAP traits, which differs on average from the native species, could have important implications for the fire regime.

·

Basic reporting

Everything looks good, all reviewer comments were adequately addressed.

Experimental design

Everything looks good now that the methods have been clarified.

Validity of the findings

Thank you for attaching the R code

Reviewer 2 ·

Basic reporting

Overall, the manuscript was improved. I have no major outstanding issues here.

Experimental design

It is still not clear exactly what material is represented by the maximum temperature measurement. There is no standard thermal properties associated with whatever object the IR gun is seeing. But that seems to be a flaw in the design of the original barbecue method (Jaureguiberry et al 2011). Can the authors make clear exactly what material is the temperature reported?

The authors did improve their description of how species were selected and what the purpose of the groups was.

Validity of the findings

I think the statistics are fine.

In regards to the authors query: "We thus do not understand the discrepancy made by the reviewer between exploratory and hypothesis testing approaches". My answer: If you run a model and report p-values you are doing hypothesis testing.

Regarding biomass/fuel load: Yes, one must include these values but often studies "correct for biomass or total fuel." I think this is dealt with adequately in this paper. The authors effectively standardize for shoot length, therefore mass differences across species arise from canopy traits.

Additional comments

My only real outstanding criticism is the authors still don't really distinguish between canopy fire/shoot flammability and litter driven fire. This does not need to be a huge issue, but the current draft confuses this. I think the authors should simply make very clear their interest in shoot flammability which is relevant given the system and also has a shorter history of study via flammability trials. This should be a relatively minor change to the main text

The authors write in their response letter that "Whether flammability pertains to the canopy or litter is only relevant in plot-scale experiments." This is not true. Past work demonstrates that the same leaf traits can have opposing effects on flammability depending on whether the fire is burning in intact live fuels or in leaf litter. Litter fire vs crown fire also represent quite different natural fire behavior and the relative importance of live fuels vs litter varies by ecosystem and fire regime (litter driven fire is less important globally but is sometimes the dominant fire type in dry forests). The authors are concerned with live fuels and that makes sense given the ecosystem. Simply be clear about this. For example, on line 100, the authors state that fine fuels "enhance flammability". This is true for live fuels but absolutely not for litter fuels where oxygen limitation dominates (eg Scarff and Westoby 2006, Magalhaes and Schwilk 2012).

The supplementary Table 1 focuses on shoot flammability studies but does not state this clearly and also throws in a couple of leaf litter flammability examples and a few studies conducted on individual leaves.
Unless the authors significantly expand this table, perhaps they should focus it only on shoot flammability trials? As it is, they include a few litter flammability trials but only studies with small scale trials that don't produce realistic flame spread rates (but include large number of species). An entire other literature is ignored. Eg work by Kreye, Varner, Schwilk or Kane in the western and eastern US, Ganteaume in France, Australian litter flammability work by Scarff and Westoby, etc. To be clear: I think it is fine to just focus on shoot flammability and canopy fire. That makes sense given the system the authors are working in. But that must be explicit. Why cite Cornwell et al and Grootemaat et al in that case, however? If those are to be included, it makes no sense to omit earlier work from which that work derived.

A few specific points by line

- lines 330-331: There are more studies than this one that show that finer particles result in reduced flammability of litterbeds as a result of oxygen limitation. The pattern is not limited to gymnosperms and was known before the study cited. See Scarff and Westoby 2006, Kane et al 2008, de Magalhaes and Schwilk 2012 for earlier studies than the Cornwell et al one highlighted. There is also all of the litter bed flammability work by Ganteaume focused on field-collected intact litter beds and going back over a decade. There is also work by western US fire ecologists that was very applied in nature that shows this result (eg Miller and Urban 1999). But oxygen limitation is a litter bed phenomenon, not really applicable to shoot flammability (Schwilk 2015).

- Line 368: I don't think one can subsume litter driven fire vs crown fire into "scale". It seems the authors did not understand my previous comments on this point.

- Line 375: Some studies have examined mixtures of species and looked at non additivity and interactions, at least for litter fuels. See de Magalhaes and Schwilk 2012, van Altena et al 2012, Gormley et al 2020, de Magalhaes and Schwilk 2021.

Some litter flammability studies. These are not necessary to cite because the authors' focus is on canopy flammability. However, the authors seem intent on including litter flammability trials in their table and discussion. I include this here as a start on this literature if the authors do want to include a table that includes both shoot and litter flammability trials.

van Altena C, van Logtestijn RSP, Cornwell WK, Cornelissen JHC. 2012. Species composition and fire: non-additive mixture effects on ground fuel flammability. Frontiers in Plant Science 3: 1–10.

Ganteaume, A., Marielle, J., Corinne, L.M., Thomas, C. and Laurent, B., 2011. Effects of vegetation type and fire regime on flammability of undisturbed litter in Southeastern France. Forest Ecology and Management, 261(12), pp.2223-2231.

Ganteaume, A., 2018. Does plant flammability differ between leaf and litter bed scale? Role of fuel characteristics and consequences for flammability assessment. International Journal of Wildland Fire, 27(5), pp.342-352.

Kane JM, Varner JM, Hiers JK. 2008. The burning characteristics of southeastern oaks: discriminating fire facilitators from fire impeders. Forest Ecology and Management 256: 2039–2045.

Kane, J.M., Kreye, J.K., Barajas-Ramirez, R. and Varner, J.M., 2021. Litter trait driven dampening of flammability following deciduous forest community shifts in eastern North America. Forest Ecology and Management, 489, p.119100.

Kane, J.M., Gallagher, M.R., Varner, J.M. and Skowronski, N.S., 2022. Evidence of local adaptation in litter flammability of a widespread fire‐adaptive pine. Journal of Ecology, 110(5), pp.1138-1148.

Kreye, J.K., Varner, J.M., Hiers, J.K. and Mola, J., 2013. Toward a mechanism for eastern North American forest mesophication: differential litter drying across 17 species. Ecological Applications, 23(8), pp.1976-1986.

Miller, C. & Urban, D. (1999) Interactions between forest heterogeneity and surface fire regimes in the southern Sierra Nevada. Canadian Journal of Forest Research, 29, 202–212.

Scarff FR, Westoby M. 2006. Leaf litter flammability in some semi-arid Australian woodlands. Functional Ecology 20: 745–752.

de Magalhaes RMQ, Schwilk DW. 2012. Leaf traits and litter flammability: evidence for non-additive mixture effects in a temperate forest. Journal of Ecology 100: 1153–1163.

An older review paper on the subject:

Varner, J.M., Kane, J.M., Kreye, J.K. and Engber, E., 2015. The flammability of forest and woodland litter: a synthesis. Current Forestry Reports, 1(2), pp.91-99.

A paper that has a short discussion of litter vs canopy fuels:

Schwilk, D.W. 2015. Dimensions of plant flammability. New Phytologist. 206:286-288

·

Basic reporting

As mentioned in the original review, this was all fine.

Experimental design

As mentioned in the original review, this was all fine

Validity of the findings

As mentioned in the original review, this is all fine

Additional comments

The authors have addressed all of my concerns. Well done.

---

## Round 0.3 · accepted · Accept

Thanks for your latest responses and revisions to the text. I support the publication of your revised manuscript.